# Predictive utility of the C-reactive protein to albumin ratio in early allograft dysfunction in living donor liver transplantation: A retrospective observational cohort study

**Jaesik Park[1], Soo Jin Lim[1], Ho Joong Choi[2], Sang Hyun Hong[1], Chul Soo Park[1], Jong Ho Choi[1], Min Suk Chae[1]***

**1** Department of Anesthesiology and Pain Medicine, Seoul St. Mary's Hospital, College of Medicine, The Catholic University of Korea, Seoul, Republic of Korea, **2** Department of Surgery, Seoul St. Mary's Hospital, College of Medicine, The Catholic University of Korea, Seoul, Republic of Korea

* shscms@gmail.com

**Data Availability Statement:** All relevant data are within the paper and its Supporting Information files.

## Abstract

### Background

This study was performed to determine the association between the ratio of C-reactive protein to albumin (CRP/ALB) and the risk of early allograft dysfunction (EAD) in patients undergoing living donor liver transplantation (LDLT).

### Patients and methods

A total of 588 adult patients undergoing LDLT were retrospectively investigated, after 22 were excluded because of signs of overt infection or history of ALB infusion. The study population was classified into high and low CRP/ALB ratio groups according to EAD. All laboratory variables, including CRP and ALB, had been collected on the day before surgery. A percentage value for the CRP/ALB ratio (%) was calculated as CRP/ALB × 100.

### Results

After LDLT, 83 patients (14.1%) suffered EAD occurrence. A higher CRP/ALB ratio was independently associated with risk of EAD, Model for End-stage Liver Disease score, fresh frozen plasma transfusion, and donor age. Based on a cutoff CRP/ALB ratio (i.e., > 20%), the probability of EAD was significantly (2-fold) higher in the high versus low CRP/ALB group. The predictive utility of CRP/ALB ratio for EAD was greater than those of other inflammatory markers. In addition, patients with a high CRP/ALB ratio had poorer survival than those with a low CRP/ALB ratio during the follow-up period.

### Conclusions

The easily calculated CRP/ALB ratio may allow estimation of the risk of EAD after LDLT and can provide additional information that may facilitate the estimation of a patient's overall condition.

**Funding:** The authors received no specific funding for this work.

**Competing interests:** The authors have declared that no competing interest exist.

**Abbreviations:** LDLT, living donor liver transplantation; ESLD, end stage liver disease; EAD, early allograft dysfunction; CRP, C-reactive protein; CRP/ALB, C-reactive protein to albumin ratio; MBP, mean blood pressure; CVP, central venous pressure; PRBC, packed red blood cell; FFP, fresh frozen plasma; SDP, single donor platelet; POD, postoperative day; MELD score, Model for end-stage liver disease score; BMI, body mass index; CRRT, continuous renal replacement therapy; NLR, neutrophil to lymphocyte ratio; HR, heart rate; IQR, interquartile.

# Introduction

Living donor liver transplantation (LDLT) has been widely accepted as one of the definitive treatments for patients with end-stage liver disease (ESLD) [1]. The patient and graft survival rates of LDLT and deceased donor liver transplantation (DDLT) are comparable [2]. However, because partial liver grafts are transplanted in LDLT, which requires concomitant liver regeneration and functional recovery, the possibility of postoperative graft dysfunction, referred to as "small-for-size syndrome," remains [3,4]. In patients with ESLD who are scheduled for liver transplantation (LT), many factors may influence the occurrence of early allograft dysfunction (EAD) after LT, such as kidney disease, renal replacement therapy, the Model for End-Stage Liver Disease (MELD) score, and donor characteristics [4–7]. Severe systemic inflammation before surgery has been shown to be related to patients' increased risk of EAD development in the LDLT setting [8,9]. A previous LDLT study demonstrated significant associations of the neutrophil-to-lymphocyte ratio (NLR) with EAD, 1-year graft failure, and mortality following surgery [8]. As the occurrence of EAD seems to be related to severe inflammatory activity derived from ischemia–reperfusion injury [10,11], the preoperative immune condition in patients undergoing LDLT may be a useful predictor of the degree of ischemia–reperfusion injury, as supported by the relationship between the preoperative levels of proinflammatory biomarkers (i.e., interleukin [IL]-2, -6, and -17) and the occurrence of EAD [9,12]. Previous studies suggested that EAD is a reversible condition that can improve during the natural postoperative course [3,6]. Nevertheless, the risk of EAD development should be estimated in patients undergoing LDLT.

Accumulating evidence supports the clinical utility of the C-reactive protein to albumin (CRP/ALB) ratio for predicting morbidity and/or mortality in critically ill patients [13,14]. CRP is a valuable acute-phase inflammatory marker that is produced in response to infection, ischemia, and trauma, and is synthesized by hepatocytes, smooth muscle cells, macrophages, endothelial cells, lymphocytes, and adipocytes under regulation by proinflammatory cytokines, particularly interleukin-6 [15–23]. A lower ALB level is related to the severity of inflammation or malnutrition [24–26], and has been proposed as a clinical tool for the estimation of hepatic insufficiency [27]. CRP and ALB have been studied widely in various clinical settings and have been recognized as valuable prognostic markers for outcomes across various diseases, including sepsis, neoplasia, critical illness after intensive care unit (ICU) admission, and hospital-acquired acute kidney injury [28–32]. The CRP/ALB ratio, a combined index of the ALB and CRP levels, is known to be related more consistently to prognosis than is CRP or ALB alone, and it may accurately reflect the degree of inflammation or nutritional deficiency [13,14,33,34]. However, as the pathological mechanisms of ESLD (i.e., hepatic insufficiency and/or malnutrition) may negatively influence CRP and ALB production, cirrhotic patients have always been excluded from previous studies [13,14,25]. Therefore, no studies have determined the associations between CRP/ALB ratio and risks of poor graft outcomes, such as EAD, in patients undergoing LDLT.

This study was performed to examine the relationship between CRP/ALB ratio and risk of EAD development in patients undergoing LDLT. In addition, we compared the utility of CRP/ALB ratio with that of other inflammatory markers for the prediction of EAD development, and sought to determine the optimal CRP/ALB ratio cutoff for predicting EAD. Postoperative outcomes were compared between the non-EAD and EAD groups.

# Patients and methods

## Ethical considerations

The Institutional Review Board of Seoul St. Mary's Hospital Ethics Committee approved the present study (KC19RESI0214) on April 15, 2019, and the study was performed in accordance

with the principles of the Declaration of Helsinki. The requirement for informed consent was waived because of the retrospective nature of the study.

## Study population

Data for 610 adult patients (aged > 19 years) who underwent elective LDLT between January 2009 and December 2018 at Seoul St. Mary's Hospital were retrospectively collected from the electronic medical record system. The exclusion criteria included signs of overt preoperative infection (e.g., pneumonia or spontaneous bacterial peritonitis), with identification of the source of infection by culture from blood, urinary ascites, and/or sputum, chest X-ray or computed tomography (CT) of the lung and/or abdomen, or based on patients' clinical presentation [35]; history of ALB infusion within 1 week before surgery (where this could artificially inflate or lower CRP/ALB ratio); and missing laboratory data. Ultimately, 588 adult patients were enrolled in this study.

## Living donor liver transplantation

The surgical and anesthetic management protocol was described in detail previously [36]. Briefly, the piggyback surgical technique was applied in the right liver lobe with reconstruction of the middle hepatic vein. After the completion of hepatic vascular and ductal anastomoses (i.e., hepatic vein, portal vein, hepatic artery, and bile duct in a serial manner), the patency of the hepatic vascular flow (i.e., portal venous flow [PVF] and hepatic arterial resistive index [HARI]) was assessed using Doppler ultrasonography (Prosound SSD-5000; Hitachi Aloka Medical, Tokyo, Japan). Surgical inflow modification via splenectomy, splenic artery ligation, or a portocaval shunt was performed based on the discretion of the transplant surgeon.

Balanced anesthesia was performed based on hemodynamic measurements (i.e., mean blood pressure [MBP] ≥ 65 mmHg and central venous pressure [CVP] ≤ 10 mmHg). According to transfusion guidelines [37], packed red blood cells (PRBCs) were transfused to achieve a hematocrit ≥ 25%, and coagulation factors (i.e., fresh frozen plasma [FFP], single donor platelets [SDPs], and cryoprecipitate) were recovered with guidance of laboratory estimation or thromboelastography.

Nephrologists routinely assessed the kidney function of patients undergoing elective LDLT according to the estimated glomerular filtration rate (eGFR), derived using the Modification of Diet in Renal Disease formula: eGFR = $175 \times$ standardized serum creatinine$^{-1.154} \times$ age$^{-0.203}$ $\times 1.212$ (if black) $\times 0.742$ (if female) [38]. Based on the eGFR, the degree of kidney function was quantified as chronic kidney disease (CKD) stage 1 (i.e., normal function and eGFR ≥ 90 mL/min/1.73 m$^2$), stage 2 (i.e., mild function loss and eGFR 60–89 mL/min/1.73 m$^2$), stage 3a (i.e., mild to moderate function loss and eGFR 45–59 mL/min/1.73 m$^2$), stage 3b (i.e., moderate to severe function loss and eGFR 30–44 mL/min/1.73 m$^2$), stage 4 (i.e., severe function loss and eGFR 15–29 mL/min/1.73 m$^2$), and stage 5 (i.e., kidney failure and eGFR < 15 mL/min/ 1.73 m$^2$) [39]. Patients with severe preoperative loss of kidney function (i.e., CKD stage 5) were provided continuous renal replacement therapy (CRRT) before and during surgery [40,41].

Significant post-reperfusion syndrome (PRS) was defined as unstable and persistent vital signs (i.e., MBP ≥ 30% in the anhepatic phase or hypotensive duration ≥ 5 min), fatal arrhythmia (i.e., asystole or ventricular tachycardia), requirement for strong rescue vasopressors (i.e., epinephrine or norepinephrine infusion), continuing or reoccurring fibrinolysis, or a requirement for anti-fibrinolytic drug treatment [42].

An immunosuppressive regimen (calcineurin inhibitor, mycophenolate mofetil, or prednisolone) was administered according to our hospital LDLT protocol. Basiliximab was

administered prior to surgery and on postoperative day (POD) 4. These immunosuppressants were gradually tapered after LDLT.

## Early allograft dysfunction

EAD was clinically determined by the presence of more than one of the following: (1) bilirubin $\geq$ 10 mg/dL on POD 7; (2) international normalized ratio [INR] $\geq$ 1.6 on POD 7; and (3) alanine or aspartate aminotransferase > 2000 IU/mL on POD 7 [5,7].

The study population was divided into non-EAD and EAD groups.

## Measurement of C-reactive protein and albumin

As part of the preoperative patient assessment, laboratory parameters, including CRP and ALB levels, were measured in all patients scheduled for LDLT. All laboratory data were collected via venous or arterial blood sampling (Clot Activator Tube/BD Vacutainer; Becton, Dickinson and Company, Franklin Lakes, NJ, USA) on the day before surgery and were measured using an automated chemistry analyzer (Hitachi 7600; Hitachi Ltd., Tokyo, Japan). If multiple tests were performed during 1 day, the results of the test performed nearest to surgery were used in the analysis, and laboratory parameters, such as CRP/ALB ratio and neutrophil to lymphocyte ratio (NLR), were derived from variables measured at the same time.

The ratio of CRP to ALB, as a percentage value, was calculated as CRP/ALB × 100 [13,14].

## Perioperative recipient and donor-graft findings

Preoperative recipient data recorded within 7 days before surgery included age, sex, body mass index (BMI), etiology of LDLT, hepatocellular carcinoma (HCC), HCC beyond the Milan criteria [43], comorbidities (i.e., diabetes mellitus and hypertension), MELD score, eGFR-based kidney function grade, hepatic decompensation (i.e., encephalopathy [West Haven grade I or II] [44], varix, and ascites) status, transthoracic echocardiography findings (i.e., ejection fraction and diastolic dysfunction [45]), and laboratory variables (i.e., white blood cell [WBC] count, NLR, CRP/ALB ratio, CRP, ALB, platelet count, sodium, potassium, calcium, glucose, and ammonia).

Intraoperative recipient findings included duration of surgery, significant PRS, vital signs (i.e., MBP, heart rate [HR], and CVP), mean lactate, blood product transfusion (i.e., PRBCs, FFP, SDPs, and cryoprecipitate), hourly fluid infusion status, hourly urine output, and Doppler ultrasonography (i.e., HARI and PVF). Donor-graft findings included age, sex, BMI, graft-recipient weight ratio, graft ischemia time, and fatty change (%).

Postoperative outcomes included total duration of hospital stay, eGFR-based kidney function grade, ascites, infection (i.e., pneumonia or sepsis), graft rejection [46], occurrence of *de novo* cancer, incidence of re-transplantation due to graft failure, and overall patient mortality.

## Statistical analysis

The perioperative recipient and donor-graft findings were compared between the non-EAD and EAD groups using the Mann–Whitney $U$ test and the $\chi^2$ test or Fisher's exact test, as appropriate. The predictive accuracy of the inflammatory markers, including CRP/ALB ratio, for EAD development were estimated using the area under the receiver operating characteristic curve (AUC). In addition, the optimal cutoff of the CRP/ALB ratio for the prediction of EAD development was determined using the AUC. The association between the perioperative factors and EAD was analyzed by univariate and multivariate logistic regression. Potentially significant factors ($p < 0.1$) in the univariate analysis were entered into multivariate forward

and backward logistic analyses. When multiple perioperative variables were inter-correlated, the most clinically relevant factors were retained in the analyses. The predictive accuracy of the models was evaluated using the AUC. The values are expressed as medians (interquartile range [IQR]) and numbers (proportion). In all analyses, $p < 0.05$ was taken to indicate statistical significance. Statistical analyses were performed using SPSS for Windows (*ver.* 24.0; SPSS Inc., Chicago, IL, USA) and MedCalc for Windows software (*ver.* 11.0; MedCalc Software, Ostend, Belgium).

## Results

### Demographic characteristics of patients undergoing LDLT

The study population was predominantly male (70.2%) and the median (IQR) age and BMI were 53 (48–59) years and 24.2 (22.1–26.6) kg/m$^2$, respectively. The most common etiologies of LDLT were as follows: hepatitis B (HBV) (54.1%), alcoholic hepatitis (23.0%), hepatitis C (6.6%), autoimmune hepatitis (4.6%), hepatitis A (4.1%), drug and toxic hepatitis (2.4%), and cryptogenic hepatitis (5.3%). The median (IQR) MELD score and CRP/ALB ratio were 15 (9–26) points and 13.9% (3.8%–54.2%), respectively. The incidence of EAD was 14.1% and the median (IQR) follow-up period was 3.6 (1.3–6.6) years.

### Comparison of pre- and intraoperative recipient and donor-graft findings according to EAD development

As shown in Table 1, there were intergroup differences in preoperative recipient findings (i.e., age; prevalence of HCC and HCC beyond the Milan criteria; MELD score; eGFR-based kidney function grade; incidence of ascites; and laboratory values for WBC count, NLR, CRP/ALB ratio, CRP, ALB, sodium, and glucose). As shown in Table 2, there were also group differences in intraoperative recipient findings (i.e., incidence of postreperfusion syndrome; average HR; total PRBC, FFP, SDP, and cryoprecipitate transfusion amounts; and hourly urine output) and donor-graft findings (i.e., age and graft ischemic time).

### Comparison of predictive accuracy for EAD development among inflammatory markers

The predictive accuracy of CRP/ALB was found to be higher than that of other inflammatory markers, such as CRP, ALB, WBC count, and NLR (Table 3). The optimal CRP/ALB ratio cut-off for prediction of EAD development was 20% (AUC: 0.722; 95% CI: 0.648–0.758; $p < 0.001$).

### Association of pre- and intraoperative findings with EAD development

In multivariate logistic regression (Table 4), CRP/ALB ratio was significantly associated with EAD development, together with the MELD score, requirement of FFP, and donor age (AUC: 0.793; 95% CI: 0.757–0.825; sensitivity: 79.5%; specificity: 71.7%; $p < 0.001$). Furthermore, the probability of EAD development in patients with a CRP/ALB ratio > 20% was 2-fold higher than in those with a CRP/ALB ratio ≤ 20% (odds ratio: 2.158; 95% CI: 1.131–4.114; $p = 0.02$; Table 4). In the EAD group (n = 83), the prevalence of patients with a CRP/ALB ratio > 20% was 75.9% (n = 63), while in the non-EAD group (n = 505), the prevalence of a CRP/ALB ratio > 20% was only 37.0% (n = 187).

**Table 1. Preoperative recipient findings of the non-EAD and EAD groups.**

| Group | non-EAD | EAD | p |
|---|---|---|---|
| n | 505 | 83 | |
| Age (years) | 54 (49–60) | 52 (46–55) | 0.015 |
| Sex (male) | 356 (70.5%) | 57 (68.7%) | 0.737 |
| Body mass index (kg/m$^2$) | | | 0.319 |
| Underweight (<18.5) | 13 (2.6%) | 5 (6.0%) | |
| Ideal weight (18.5–24.9) | 283 (56.0%) | 42 (50.6%) | |
| Overweight (25–29.9) | 165 (32.7%) | 27 (32.5%) | |
| Obese (≥30) | 44 (8.7%) | 9 (10.8%) | |
| *Etiology* | | | 0.605 |
| Alcohol | 120 (23.8%) | 15 (18.1%) | |
| Hepatitis A | 19 (3.8%) | 5 (6.0%) | |
| Hepatitis B | 273 (54.1%) | 45 (54.2%) | |
| Hepatitis C | 33 (6.5%) | 6 (7.2%) | |
| Autoimmune | 21 (4.2%) | 6 (7.2%) | |
| Drug & Toxin | 11 (2.2%) | 3 (3.6%) | |
| Cryptogenic | 28 (5.5%) | 3 (3.6%) | |
| Hepatocellular carcinoma | 238 (47.1%) | 18 (21.7%) | <0.001 |
| Beyond Milan criteria | 7 (2.9%) | 9 (50.0%) | <0.001 |
| *Comorbidity* | | | |
| Diabetes mellitus | 132 (26.1%) | 21 (25.3%) | 0.872 |
| Hypertension | 104 (20.6%) | 13 (15.7%) | 0.297 |
| MELD score (point) | 13 (8–22) | 28 (16–38) | < 0.001 |
| *eGFR based kidney function grade* | | | 0.001 |
| Stage 1 | 246 (48.7%) | 31 (37.3%) | |
| Stage 2 | 110 (21.8%) | 12 (14.5%) | |
| Stage 3a | 34 (6.7%) | 4 (4.8%) | |
| Stage 3b | 25 (5.0%) | 11 (13.3%) | |
| Stage 4 | 28 (5.5%) | 4 (4.8%) | |
| Stage 5 | 62 (12.3%) | 21 (25.3%) | |
| *Hepatic decompensation* | | | |
| Encephalopathy (West Haven grade I or II) | 43 (8.5%) | 11 (13.3%) | 0.166 |
| Varix | 124 (24.6%) | 20 (24.1%) | 0.928 |
| Ascites | 227 (45.0%) | 50 (60.2%) | 0.010 |
| *Transthoracic echocardiography* | | | |
| Ejection fraction (%) | 65 (62–67) | 65 (63–67) | 0.310 |
| Diastolic dysfunction | 221 (43.8%) | 33 (39.8%) | 0.495 |
| *Laboratory variables* | | | |
| Hemoglobin (g/dL) | 9.8 (8.4–11.7) | 9.0 (8.1–11.3) | 0.056 |
| WBC count (× 10$^9$/L) | 4.1 (2.7–6.2) | 7.8 (3.8–12.0) | <0.001 |
| Neutrophil to lymphocyte ratio (%) | 244.3 (153.0–505.2) | 595.1 (301.0–1044.6) | <0.001 |
| C-reactive protein to albumin ratio (%) | 10.3 (3.2–42.5) | 55.0 (20.9–134.1) | <0.001 |
| C-reactive protein (mg/dL) | 0.3 (0.1–1.2) | 1.5 (0.7–4.0) | <0.001 |
| Albumin (g/dL) | 3.0 (2.7–3.5) | 2.8 (2.5–2.9) | <0.001 |
| Platelet count (× 10$^9$/L) | 63.0 (46.0–103.0) | 62.0 (39.0–87.0) | 0.285 |
| Sodium (mEq/L) | 139.0 (135.0–142.0) | 138.0 (133.0–141.0) | 0.023 |
| Potassium (mEq/L) | 4.0 (3.7–4.3) | 3.9 (3.5–4.3) | 0.182 |
| Calcium (mg/dL) | 8.4 (8.0–8.8) | 8.4 (7.9–8.8) | 0.504 |

*(Continued)*

**Table 1.** (Continued)

| Group | non-EAD | EAD | *p* |
|---|---|---|---|
| n | 505 | 83 | |
| Glucose (mg/dL) | 108.0 (92.0–136.5) | 123.0 (96.0–156.0) | 0.035 |
| Ammonia (μg/dL) | 98.0 (66.0–153.5) | 93.0 (67.0–149.0) | 0.635 |

**Abbreviations:** EAD, early allograft dysfunction; eGFR, estimated glomerular filtration rate; CRRT, continuous renal replacement therapy; MELD, Model for End-stage Liver Disease; WBC, white blood cell

**NOTE:** Values are medians (interquartile range) or numbers (proportion).

## Comparison of postoperative outcomes between patients with and without EAD

Patients with EAD had a longer duration of hospitalization and higher rates of poor kidney function, infection, and re-transplantation due to graft failure, as well as higher overall mortality, than those without EAD (Table 5). Patients with a CRP/ALB ratio > 20% had a longer duration of hospitalization and higher rates of poor kidney function, infection, and re-

**Table 2.** Intraoperative recipient and donor-graft findings of the non-EAD and EAD groups.

| Group | non-EAD | EAD | *p* |
|---|---|---|---|
| n | 505 | 83 | |
| *Intraoperative recipient findings* | | | |
| Surgical duration (min) | 500 (450–565) | 510 (470–590) | 0.213 |
| Postreperfusion syndrome | 108 (21.4%) | 29 (34.9%) | 0.007 |
| *Average of vital signs* | | | |
| MBP (mmHg) | 74 (69–80) | 75 (65–81) | 0.747 |
| HR (beats/min) | 89 (80–99) | 94 (83–107) | 0.013 |
| CVP (mmHg) | 9 (7–11) | 9 (7–12) | 0.548 |
| Mean lactate (mmol/L) | 3.7 (2.9–4.9) | 3.8 (2.6–5.5) | 0.855 |
| *Blood product transfusion (unit)* | | | |
| Packed red blood cells | 7 (4–13) | 11 (8–16) | < 0.001 |
| Fresh frozen plasma | 7 (4–10) | 10 (8–15) | < 0.001 |
| Single donor platelet | 1 (0–2) | 1 (0–2) | 0.002 |
| Cryoprecipitate | 0 (0–0) | 0 (0–0) | 0.017 |
| Hourly fluid infusion (mL/kg/h) | 9.6 (6.7–12.9) | 9.5 (6.4–13.6) | 0.645 |
| Hourly urine output (mL/kg/h) | 1.3 (0.7–2.1) | 1.0 (0.3–1.7) | 0.001 |
| Doppler ultrasonography | | | |
| Hepatic arterial resistive index | 0.64 (0.63–0.66) | 0.64 (0.64–0.67) | 0.936 |
| Portal venous flow (mL/min) | 2180.2 (1749.5–2281.3) | 2179.1 (1647.0–2183.1) | 0.394 |
| *Donor-graft findings* | | | |
| Age (years) | 35 (26–40) | 35 (31–46) | 0.017 |
| Sex (male) | 344 (68.1%) | 52 (62.7%) | 0.325 |
| Body mass index (kg/m$^2$) | 23.8 (21.8–25.4) | 23.8 (22.3–25.2) | 0.990 |
| GRWR (%) | 1.2 (1.0–1.5) | 1.2 (1.0–1.5) | 0.964 |
| Graft ischemic time (min) | 93 (69–122) | 113 (74–192) | 0.002 |
| Fatty change (%) | 4.7 (1.0–5.0) | 4.7 (1.0–5.0) | 0.341 |

**Abbreviations:** EAD, early allograft dysfunction; GRWR, graft recipient weight ratio; MBP, mean blood pressure; HR, heart rate; CVP, central venous pressure

**NOTE:** Values are medians (interquartile range) or numbers (proportion).

**Table 3. Predictive accuracy of inflammatory markers for EAD development.**

|  | AUC | 95% CI | *p* |
|---|---|---|---|
| C-reactive protein to albumin ratio (%) | 0.722 | 0.684–0.758 | < 0.001 |
| C-reactive protein (mg/dL) | 0.711 | 0.672–0.747 | < 0.001 |
| Albumin (g/dL) | 0.670 | 0.630–0.708 | < 0.001 |
| WBC count (× 10$^9$/L) | 0.672 | 0.632–0.710 | < 0.001 |
| Neutrophil to lymphocyte ratio (%) | 0.717 | 0.679–0.753 | < 0.001 |

**Abbreviations:** AUC, area under the curve; WBC, white blood cell

transplantation due to graft failure, as well as higher overall mortality, than those with a CRP/ALB ratio ≤ 20% (S1 Table).

## Discussion

The main findings of our study were that a higher CRP/ALB ratio was independently associated with the risk of EAD development after LDLT, together with the MELD score, requirement for FFP transfusion, and donor age. Based on the CRP/ALB ratio cutoff of (> 20%) for EAD, the probability of EAD development was significantly (2-fold) higher in patients with a CRP/ALB ratio > 20% than in those with a ratio ≤ 20%. Compared to other inflammatory markers, the predictive utility of the CRP/ALB ratio for EAD development was higher.

EAD occurs in about 20% of patients who undergo LT [5,7], and 14.1% of our patients experienced this complication. EAD pathogenesis is potentially related to inflammatory and oxidative stress in the transplanted graft in response to ischemia–reperfusion injury [9,10,12]. Many studies have reported that EAD development negatively affected postoperative outcomes, resulting in short- and long-term kidney dysfunction, prolonged hospital stay, early liver graft loss, and poor patient survival [5–7,47]. Our results agree with these findings, in that patients with EAD had higher incidences of poor kidney function and infection, longer hospital stays, and re-transplantation due to graft failure, and worse overall patient mortality rate than those without EAD. A previous study by Wadei *et al.*[47] indicated an association of EAD occurrence with new acute kidney injury requiring renal replacement therapy within the first month and end-stage renal disease within the first year after LT. They speculated that the development of EAD contributed to an increase in oxidative stress due to the production of reactive oxygen species and triggered a systemic inflammatory response that expanded beyond the liver graft.

In critically ill patients, CRP/ALB ratio reflects both inflammatory activation and nutritional deficiency [48,49], and has gradually been accepted as an emerging prognostic marker of morbidity and mortality [13,14]. CRP level increases during complex and stressful surgeries, as well as in numerous diseases, such as sepsis, decompensated heart failure, and cerebral disease [19–21,50]. In patients with chronic liver disease [51–57], a higher serum CRP concentration serves as an independent prognostic marker related to morbidity and mortality. In a study of *Escherichia coli* infection, the host capacity for CRP production was maintained in cirrhotic patients with severe liver dysfunction [58]. Another infection study suggested that serum levels of acute-phase proteins (CRP and procalcitonin) originating from the liver were comparable between patients with and without cirrhosis [59]. Recent studies have suggested that CRP can serve as a surrogate biomarker for acute or chronic systemic inflammation, which may be poorly predicted by the MELD score in patients with ESLD, such as alcoholic hepatitis, hepatocellular carcinoma, tissue necrosis, and bacterial translocation. Furthermore, increased CRP was significantly associated with patient outcomes such as hepatic insult and/or extrahepatic

**Table 4. Association of pre- and intraoperative recipient and donor-graft findings with EAD development.**

| | Univariable logistic regression analysis | | | | Multivariable logistic regression analysis | | | |
|---|---|---|---|---|---|---|---|---|
| | $\beta$ | Odds ratio | 95% CI | *p* | $\beta$ | Odds ratio | 95% CI | *p* |
| *Preoperative recipient finding* | | | | | | | | |
| Age (years) | -0.022 | 0.978 | 0.953–1.005 | 0.106 | | | | |
| Sex (male *vs.* female) | 0.086 | 1.090 | 0.660–1.800 | 0.737 | | | | |
| Body mass index (kg/m$^2$) | -0.026 | 0.974 | 0.915–1.037 | 0.409 | | | | |
| *Comorbidity* | | | | | | | | |
| Diabetes mellitus | -0.044 | 0.957 | 0.562–1.631 | 0.872 | | | | |
| Hypertension | -0.334 | 0.716 | 0.381–1.344 | 0.299 | | | | |
| MELD score (point) | 0.083 | 1.087 | 1.064–1.110 | <0.001 | 0.063 | 1.065 | 1.040–1.090 | <0.001 |
| eGFR based kidney function grade | 0.206 | 1.229 | 1.093–1.381 | 0.001 | | | | |
| *Hepatic decompensation* | | | | | | | | |
| Encephalopathy | 0.496 | 1.641 | 0.809–3.330 | 0.170 | | | | |
| Varix | -0.025 | 0.975 | 0.567–1.678 | 0.928 | | | | |
| Ascites | 0.618 | 1.856 | 1.156–2.979 | 0.010 | | | | |
| *Transthoracic echocardiography* | | | | | | | | |
| Ejection fraction (%) | 0.031 | 1.031 | 0.977–1.088 | 0.262 | | | | |
| Diastolic dysfunction | -0.165 | 0.848 | 0.528–1.362 | 0.495 | | | | |
| *Laboratory variables* | | | | | | | | |
| Hemoglobin (g/dL) | -0.095 | 0.909 | 0.815–1.015 | 0.090 | | | | |
| WBC count (x 10$^9$/L) | 0.076 | 1.079 | 1.039–1.120 | <0.001 | | | | |
| Neutrophil to lymphocyte ratio (%) | 0.056 | 1.057 | 1.025–1.091 | 0.001 | | | | |
| CRP/ALB (%) (continuous) | 0.008 | 1.008 | 1.005–1.011 | <0.001 | 0.004 | 1.004 | 1.001–1.007 | 0.004 |
| CRP/ALB (%) (dichotomized)* | | | | | | | | |
| ≤ 20% | reference | | | | | | | |
| > 20% | 1.678 | 5.357 | 3.139–9.142 | <0.001 | 0.769 | 2.158 | 1.131–4.114 | 0.020 |
| C-reactive protein (mg/dL) | 0.192 | 1.211 | 1.101–1.332 | <0.001 | | | | |
| Albumin (g/dL) | -1.160 | 0.313 | 0.195–0.504 | <0.001 | | | | |
| Platelet count (x 10$^9$/L) | -0.002 | 0.998 | 0.993–1.002 | 0.313 | | | | |
| Sodium (mEq/L) | -0.045 | 0.956 | 0.917–0.996 | 0.032 | | | | |
| Potassium (mEq/L) | -0.196 | 0.822 | 0.554–1.219 | 0.329 | | | | |
| Calcium (mg/dL) | -0.031 | 0.970 | 0.737–1.276 | 0.826 | | | | |
| Glucose (mg/dL) | 0.002 | 1.002 | 0.998–1.005 | 0.368 | | | | |
| Ammonia (μg/dL) | -0.001 | 0.999 | 0.996–1.002 | 0.567 | | | | |
| *Intraoperative recipient finding* | | | | | | | | |
| Surgical duration (min) | 0.001 | 1.001 | 0.999–1.004 | 0.291 | | | | |
| Postreperfusion syndrome | 0.680 | 1.974 | 1.199–3.251 | 0.008 | | | | |
| Average of vital signs | | | | | | | | |
| MBP (mmHg) | -0.002 | 0.998 | 0.973–1.024 | 0.871 | | | | |
| HR (beats/min) | 0.022 | 1.023 | 1.006–1.039 | 0.007 | | | | |
| CVP (mmHg) | 0.029 | 1.029 | 0.958–1.106 | 0.436 | | | | |
| Mean lactate (mmol/L) | 0.055 | 1.056 | 0.997–1.119 | 0.062 | | | | |
| *Blood product transfusion (unit)* | | | | | | | | |
| Packed red blood cell | 0.039 | 1.040 | 1.017–1.063 | 0.001 | | | | |
| Fresh frozen plasma | 0.067 | 1.069 | 1.038–1.101 | <0.001 | 0.041 | 1.042 | 1.007–1.078 | 0.019 |
| Single donor platelet | 0.077 | 1.080 | 1.000–1.167 | 0.050 | | | | |
| Cryoprecipitate | 0.094 | 1.099 | 1.004–1.203 | 0.042 | | | | |
| Hourly fluid infusion (mL/kg/h) | -0.001 | 0.999 | 0.975–1.024 | 0.966 | | | | |

(*Continued*)

**Table 4.** (Continued)

| | Univariable logistic regression analysis | | | | Multivariable logistic regression analysis | | | |
|---|---|---|---|---|---|---|---|---|
| | β | Odds ratio | 95% CI | p | β | Odds ratio | 95% CI | p |
| Hourly urine output (mL/kg/h) | -0.364 | 0.695 | 0.541–0.894 | 0.005 | | | | |
| Doppler ultrasonography | | | | | | | | |
| Hepatic arterial resistive index | -0.140 | 0.869 | 0.024–31.114 | 0.939 | | | | |
| Portal venous flow (mL/min) | 0.000 | 1.000 | 1.000–1.000 | 0.931 | | | | |
| *Donor-graft finding* | | | | | | | | |
| Age (years) | 0.026 | 1.027 | 1.006–1.047 | 0.009 | 0.023 | 1.023 | 1.001–1.047 | 0.041 |
| Sex(male) | 0.242 | 1.274 | 0.786–2.064 | 0.326 | | | | |
| Body mass index (kg/m$^2$) | -0.016 | 0.984 | 0.909–1.064 | 0.682 | | | | |
| GRWR (%) | 0.337 | 1.401 | 0.770–2.548 | 0.270 | | | | |
| Graft ischemic time (min) | 0.004 | 1.004 | 1.002–1.006 | <0.001 | | | | |
| Fatty change (%) | 0.003 | 1.003 | 0.969–1.038 | 0.865 | | | | |

Odds ratio of CRP/ALB (%) (dichotomized)* was a value from another univariable and multivariable logistic regression model without CRP/ALB (continuous).

**Abbreviations:** EAD, early allograft dysfunction; MELD, model for end-stage liver disease; eGFR, estimated glomerular filtration rate; CRRT, continuous renal replacement therapy; CRP/ALB, C-reactive protein to albumin ratio; GRWR, graft-recipient-weight-ratio; MBP, mean blood pressure; HR, heart rate; CVP, central venous pressure

organ dysfunction [56,60]. Although CRP is largely synthesized in the liver, other cells (smooth muscle cells, macrophages, endothelial cells, lymphocytes, and adipocytes) and reserved hepatocytes seem to be able to produce CRP in response to increasing IL-6, which is continuously activated by lipopolysaccharide-binding protein during cirrhosis [61–63]. Additionally, CRP plays critical roles in the host response to infection and inflammation, such as the complement pathway, recruitment of leukocytes, facilitation of apoptosis and phagocytosis, and production of nitric oxide and cytokines [23].

**Table 5.** Postoperative outcomes of the non-EAD and EAD groups.

| Group | non-EAD | EAD | p |
|---|---|---|---|
| n | 505 | 83 | |
| Hospital stay (day) | 25 (21–35) | 31 (22–54) | 0.002 |
| During follow-up period | | | |
| eGFR based kidney function grade | | | <0.001 |
| Stage 1 | 365 (72.7%) | 31 (40.3%) | |
| Stage 2 | 76 (15.1%) | 15 (19.5%) | |
| Stage 3a | 24 (4.8%) | 9 (11.7%) | |
| Stage 3b | 26 (5.2%) | 14 (18.2%) | |
| Stage 4 | 8 (1.6%) | 5 (6.5%) | |
| Stage 5 | 3 (0.6%) | 3 (3.9%) | |
| Ascites | 171 (33.9%) | 24 (28.9%) | 0.375 |
| Infection | 36 (7.1%) | 20 (24.1%) | <0.001 |
| Graft rejection | 99 (19.6%) | 17 (20.5%) | 0.852 |
| *de novo* Cancer occurrence | 38 (7.5%) | 6 (7.2%) | 0.924 |
| Re-transplantation | 12 (2.4%) | 12 (14.5%) | <0.001 |
| Overall patient mortality | 58 (11.5%) | 30 (36.1%) | <0.001 |

**Abbreviations:** EAD, early allograft dysfunction; eGFR, estimated glomerular filtration rate
**NOTE:** Values are expressed as median (interquartile) and number (proportion).

Hypoalbuminemia is commonly seen in septic patients and is associated with the severity of sepsis [64]. A decrease in serum ALB level of 10 g/L is associated with increased risks of mortality and morbidity, and prolonged ICU and hospital stays [65]. Circulating albumin has been considered as an estimation tool for hepatic insufficiency as a component of the Child-Pugh classification [27]. Additionally, because circulating albumin plays a role in scavenging and disposing of inflammatory mediators such as cytokines and lipopolysaccharide, a low albumin level may contribute to the aggravation of inflammatory activation [66]. Compared to other more well-established indicators of inflammation (e.g., the Glasgow Prognostic Score [GPS], modified GPS, NLR, platelet-lymphocyte ratio, prognostic index, and prognostic nutritional index), CRP/ALB ratio was a better predictor of poor overall patient survival [67]. However, a study of ICU patients suggested that the predictive accuracy of CRP/ALB ratio for 30-day mortality did not exceed that of the ALB level alone, the Acute Physiologic Assessment and Chronic Health Evaluation II, or the Charlson Comorbidity Index [68]. However, because patients admitted to the ICU due to mixed and heterogenous causes were enrolled in the study by Oh *et al.* [68], the results may have limited relevance for specific cases such as cirrhosis, malignancy, or infection.

Our results are supported by those from previous studies of chronic liver disease [51,53] showing that an increased CRP/ALB ratio was significantly associated with the severity of hepatic inflammation and prognosis. Our patients with EAD showed higher levels of inflammatory markers (i.e., WBC count, NLR, CRP/ALB ratio, CRP, and ALB) than those without EAD. In a multivariate regression analysis, a higher CRP/ALB ratio remained an independent predictor of EAD occurrence after LDLT, and the AUC of the CRP/ALB ratio was higher than other inflammatory indicators, such as NLR. Therefore, the predictive utility of the CRP/ALB ratio for EAD development is likely to be high, and the magnitude of the CRP/ALB ratio may be related to the inflammatory and metabolic demand of the liver graft, and thus the risk of EAD development. Additionally, previous studies have suggested that antiviral treatment leads to significant amelioration of the MELD score and an improved portal hypertension-related prognosis in patients with HBV-related decompensated cirrhosis [69,70]. An HBV study by Huang *et al.*[53] found that the CRP/ALB ratio decreased significantly in response to antiviral therapy. This suggests that antiviral therapy influences the expected capacity of the CRP/ALB ratio, which should be confirmed in future studies investigating the dose and timing of antiviral therapy in other types of hepatitis.

Our study, supported by Olthoff *et al.*[7], suggests that MELD score and donor age are associated significantly with the risk of EAD occurrence within the first week after LT. Poor implanted liver graft function may contribute to the persistence of a preoperatively high MELD score, including alterations of the total bilirubin level, international normalized ratio, and creatinine level, until postoperative day 7. In addition, donor age seems to have an impact on the initiation of the molecular repair pathway, and solid (i.e., liver, lung, heart, and kidney) grafts from older donors exhibit prolonged functional recovery more frequently than do those from younger donors [71]. In patients undergoing LDLT, grafts from younger donors have better vascular resistance, higher compliance of the hepatic parenchyma, and more favorable regeneration than those from older donors. Donor age is thus a critical and independent factor in recipient survival [72,73].

In our study, a greater requirement for FFP transfusion was a prognostic factor for the occurrence of EAD. Wadei *et al.*[47] reported that patients with EAD required greater amounts of blood product (including FFP) transfusion than did those without EAD. As persistent uncontrolled coagulopathy be an early sign of transplanted graft dysfunction [74], it is not surprising that the requirement for FFP transfusion was higher in the EAD group than in the

non-EAD group. Poor liver graft recovery seems to begin with severe ischemia–reperfusion injury, followed by the continuous presence of coagulopathy [75].

This study has some limitations. First, we were not able to clarify the underlying mechanisms. Although both CRP and ALB reflect systemic inflammation, further prospective studies are required to elucidate the pathophysiological processes underlying the association of the CRP/ALB ratio with EAD. As albumin has a positive impact on homeostasis, including the stabilization of endothelial glycocalyx [76], the development of a clinical algorithm may advance therapeutic practice. Second, there are characteristic differences between LT from living and deceased donors. LDLT allografts are partial livers from healthy donors (i.e., smaller but with a healthy status), while DDLT allografts are whole livers from deceased donors with poorer clinical conditions and prolonged graft cold ischemic times (i.e., larger but with some injury) [5,7]. Therefore, the impact of donor-graft factors on the development of EAD may have been reduced in our LDLT study. Additional studies are required to validate the predictive role of the CRP/ALB ratio for EAD in LT not only from living donors, but also from deceased donors. Third, the incidence of EAD was lower in patients with than in those without HCC, but higher in the HCC beyond the Milan criteria group than in the HCC within the Milan criteria group. As this finding may reflect selection bias, we did not analyze the association of HCC with EAD occurrence. Further studies are required to investigate the associations only in patients with HCC undergoing LDLT. Fourth, because of the retrospective study design, not all potential confounding factors could be removed. Finally, although clinical factors related to EAD were controlled for in multivariate analysis, the possibility of selection bias could not be completely excluded, and the results should be interpreted carefully.

## Conclusions

Our study showed that CRP/ALB ratio, measured prior to LDLT, was an independent predictor of EAD development, together with a high MELD score, greater requirement for FFP transfusion, and older donor age. Therefore, the easily calculated CRP/ALB ratio may allow estimation of the risk of EAD after LDLT and can provide additional information that may facilitate the overall estimation of a patient's condition.

## Supporting information

**S1 Table. Postoperative outcomes of patients with CRP/ALB ratios $\leq$ and $>$ 20%.**
(DOCX)

## Acknowledgments

All authors thank Suna Yu and Hyeji An (Anesthesia Nursing Unit, Seoul St. Mary's Hospital, College of Medicine, The Catholic University of Korea, Seoul, Republic of Korea) for participation of our study.

## Author Contributions

**Conceptualization:** Jaesik Park, Min Suk Chae.

**Data curation:** Jaesik Park, Soo Jin Lim, Ho Joong Choi, Sang Hyun Hong, Chul Soo Park, Jong Ho Choi, Min Suk Chae.

**Formal analysis:** Jaesik Park, Soo Jin Lim, Ho Joong Choi, Sang Hyun Hong, Chul Soo Park, Jong Ho Choi, Min Suk Chae.

**Investigation:** Jaesik Park, Soo Jin Lim, Ho Joong Choi, Sang Hyun Hong, Chul Soo Park, Jong Ho Choi, Min Suk Chae.

**Methodology:** Min Suk Chae.

**Supervision:** Min Suk Chae.

**Visualization:** Min Suk Chae.

**Writing – original draft:** Jaesik Park, Min Suk Chae.

**Writing – review & editing:** Min Suk Chae.

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
