## [Decision Letter · Decision Letter 0]

16 Sep 2019

PONE-D-19-22048

Predictive utility of the C-reactive protein to albumin ratio in early allograft dysfunction in living donor liver transplantation: A retrospective observational cohort study

PLOS ONE

Dear Dr. Chae,

Thank you for submitting your manuscript to PLOS ONE. After careful consideration, we feel that it has merit but does not fully meet PLOS ONE’s publication criteria as it currently stands. Therefore, we invite you to submit a revised version of the manuscript that addresses the points raised during the review process.

The reviewers and I believe that this work is a valuable and important one in this field. However, many concerned issues should be addressed and improved clearly before acceptance. Please refer the following comments in detail from two experts.

We would appreciate receiving your revised manuscript by Oct 31 2019 11:59PM. To enhance the reproducibility of your results, we recommend that if applicable you deposit your laboratory protocols in protocols.io, where a protocol can be assigned its own identifier (DOI) such that it can be cited independently in the future. For instructions see: http://journals.plos.org/plosone/s/submission-guidelines#loc-laboratory-protocols

We look forward to receiving your revised manuscript.

Kind regards,

Yun-Wen Zheng

Academic Editor

PLOS ONE

Journal Requirements:

Reviewers' comments:

Reviewer's Responses to Questions

**Comments to the Author**

1. Is the manuscript technically sound, and do the data support the conclusions?

Reviewer #1: Partly

Reviewer #2: Yes

2. Has the statistical analysis been performed appropriately and rigorously? 

Reviewer #1: N/A

Reviewer #2: Yes

3. Have the authors made all data underlying the findings in their manuscript fully available?

Reviewer #1: Yes

Reviewer #2: Yes

4. Is the manuscript presented in an intelligible fashion and written in standard English?

Reviewer #1: Yes

Reviewer #2: Yes

5. Review Comments to the Author

Reviewer #1: Chae et al. evaluated relationship between preoperative CRP/Alb and EAD after LDLT in their study population of 588 patients. They find patients with CRP/Alb >20% showed 2-fold higher incidence of EAD. Further, patients with higher CRP/Alb showed poorer overall survival than those with lower CRP/Alb. They concluded higher CRP/alb could predict EAD after LDLT. We think this result is somehow understandable. However, we found some issues.

major

1. First of all, CRP and Alb are both generated in the liver. Can we simply trust the values of CRP as an inflammatory parameter in patients with ESLD? The authors should explain this point.

2. This is two stage story (high CRP/alb=>EAD=>graft failure). Although the authors showed correlation between high CRP/alb and EAD by multivariate analysis, they did not prove correlation between EAD and graft failure correctly because the graph they showed us was simple univariate analysis (Fig 1). We found significantly higher pre-operative risks in EAD group (background disease, MELD score, ischemia time). The same thing in Fig 2. If they want to show this result, they should perform propensity score matching showing higher CRP/alb could lead to graft failure or EAD. Also, how was the results of Cox-regression analysis, using CRP/Alb for graft failure?

3. The authors focus on EAD. Why did the authors showed overall survival? not graft survival in Fig 1 and 2?

4. Why could the authors find donor factors being the risk for EAD? We could not simply accept only one recipient preoperative parameter can be indicator for EAD and graft loss (including long-term patient survival). They should refer this point at discussion, not only one sentence as a limitation since many previous articles focused on donor factors in LDLT.

5. As an objective parameter, “intraoperative portal pressure (or portal flow) after reperfusion “ is very important since it is well-known factor for developing “small for size syndrome”.

6. In addition to EAD, how was the postoperative complications, such as ascites, infection, etc (makers for small for size syndrome) in high CRP/alb ?

7. Since CRP/alb is preoperative parameter, if they find CRP/alb>20, what do they do? Do they treat them with antibiotics? Have the patinet better nutritional condition before preceding to transplant? They should discuss this point.

Minor

1. Table 1, they should stratify BMI because we don’t believe BMI will show liner relationship with EAD.

Reviewer #2: This study was performed to determine the association between the ratio of C-reactive protein to albumin (CRP/ALB) and the risk of early allograft dysfunction(EAD) in patients undergoing living donor liver transplantation (LDLT).The easily calculated CRP/ALB ratio may allow estimation of the risk of

EAD after LDLT and can provide additional information that may facilitate overall estimation of the patient’s condition.

The manuscript is written well but there are several concerns.

comments.

1. In table 3, The predictive accuracy of CRP/ALB was found to be higher than that of other inflammatory

markers, such as CRP, ALB, WBC count, and NLR. But in table 4, among inflammatory markers, CRP only nor ALB are not listed. The authors should list up CRP and ALB independently.

2. In table 5, Patients with EAD had a longer duration of hospitalization and higher incidence rates of poor kidney function and infection than those without EAD. These postoperative conditions such as poor kidney function and infection cause EAD? If these postoperative parameters included in multivariable logistic regression, how about significancy?

6. PLOS authors have the option to publish the peer review history of their article (what does this mean?). If published, this will include your full peer review and any attached files.

Reviewer #1: No

Reviewer #2: Yes: Soichiro Murata

---

## [Author Response · Author response to Decision Letter 0]

2 Nov 2019

A point-by-point response letter

Yun-Wen Zheng

Academic Editor

PLoS ONE

Dear Dr. Zheng:

We are pleased to have had the opportunity to revise and resubmit our manuscript. We would like to thank the reviewers for their insightful comments, which have helped to significantly improve our manuscript. Our point-by-point responses to each comment are presented below. Our revisions and responses are provided directly after each comment. The revised parts of the manuscript are shown using tracked changes.

The English in this document has been checked by at least two professional editors, both native speakers of English. For a certificate, please see:

http://www.textcheck.com/certificate/E7niGB

Reviewers’ comments

Reviewer #1: Chae et al. evaluated relationship between preoperative CRP/Alb and EAD after LDLT in their study population of 588 patients. They find patients with CRP/Alb >20% showed 2-fold higher incidence of EAD. Further, patients with higher CRP/Alb showed poorer overall survival than those with lower CRP/Alb. They concluded higher CRP/alb could predict EAD after LDLT. We think this result is somehow understandable. However, we found some issues.

major

1. First of all, CRP and Alb are both generated in the liver. Can we simply trust the values of CRP as an inflammatory parameter in patients with ESLD? The authors should explain this point.

Response: The outcome and reversibility of hepatic decompensation vary according to the nature and severity of the acute hepatic insult, and according to the degree of dysfunction of extrahepatic organ systems. Recent studies have shown that, in acutely ill patients with cirrhosis, systemic inflammatory response syndrome (SIRS) with or without a bacterial infection is an independent predictor of survival and is associated with the development of portal hypertension-related complications. Liver function appears not be the main determinant of outcome in patients with cirrhosis who experience multiorgan failure. Therefore, the negative impact of systemic inflammation may be poorly predicted by MELD scores. Conventional parameters for diagnosing SIRS lack sensitivity and specificity in patients with advanced cirrhosis because of hypersplenism, hyperventilation associated with encephalopathy, hyperkinetic circulation, or the use of beta-blockers. 

Previous studies suggested that high CRP levels are often observed in patients with advanced liver failure. CRP can be used as a surrogate biomarker for acute or chronic systemic inflammation in patients with ESLD, such as alcoholic hepatitis, hepatocellular carcinoma, tissue necrosis, and bacterial translocation, and high CRP is significantly associated with patient outcome. Although CRP is largely synthesized by the liver, other cells (e.g., smooth muscle cells, macrophages, endothelial cells, lymphocytes, and adipocytes) and reserved hepatocytes seem to be able to effectively and promptly produce CRP as an acute-phase protein, mainly in response to increasing interleukin-6, which is continuously activated by the lipopolysaccharide-binding protein during cirrhosis. Hence, we consider measuring CRP to be a simple and accurate way of diagnosing systemic inflammation and providing an indication of prognosis in cirrhotic patients.

A decrease in serum albumin level is associated with a high risk of patient morbidity and mortality. Circulating albumin has been used to estimate hepatic insufficiency as a component of the Child-Pugh classification. Additionally, because circulating albumin plays a role in scavenging and disposing of inflammatory mediators, such as cytokines and lipopolysaccharides, a low albumin level may contribute to the aggravation of inflammatory activation. 

Our results are supported by previous studies of chronic liver disease showing that a higher CRP/ALB ratio is significantly associated with the severity of hepatic inflammation and prognosis. Our patients with EAD showed higher levels of inflammatory markers (e.g., WBC count, NLR, CRP/ALB ratio and CRP) than those without EAD. In a multivariate regression analysis, increased CRP/ALB ratio remained an independent predictor of EAD occurrence after LDLT, and the AUC of the CRP/ALB ratio was higher than other inflammatory indicators, such as NLR. Therefore, the predictive utility of the CRP/ALB ratio for EAD development is likely to be high, and the magnitude of the CRP/ALB ratio may be related to the inflammatory and metabolic demand of the liver graft, and thus the risk of EAD (Discussion; pages 28 - 29). 

2. This is two stage story (high CRP/alb=>EAD=>graft failure). Although the authors showed correlation between high CRP/alb and EAD by multivariate analysis, they did not prove correlation between EAD and graft failure correctly because the graph they showed us was simple univariate analysis (Fig 1). We found significantly higher pre-operative risks in EAD group (background disease, MELD score, ischemia time). The same thing in Fig 2. If they want to show this result, they should perform propensity score matching showing higher CRP/alb could lead to graft failure or EAD. Also, how was the results of Cox-regression analysis, using CRP/Alb for graft failure?

Response: We compared the overall survival rates between patients with and without EAD occurrence (Fig. 1) and between patients with CRP/ALB ratios ≤ 20% and > 20% (Fig. 2) using a univariate analysis (Kaplan-Meier analysis with the log-rank test). The primary aim of our study was to investigate the association of perioperative clinical factors, including CRP/ALB ratio, with EAD occurrence, and additionally to compare postoperative outcomes, including overall patient survival, between patients with and without EAD. The analysis of risk factors for overall survival, including CRP/ALB ratio, is beyond the scope of this study. Therefore, we deleted the redundant Figures 1 and 2 and added the incidence of re-transplantation due to graft failure and overall patient mortality using the χ2 test or Fisher’s exact test, as appropriate, in Table 5 (pages 25 - 26). In future, we will investigate the predictive role of the CRP/ALB ratio in patient or graft survival after LDLT. 

3. The authors focus on EAD. Why did the authors showed overall survival? not graft survival in Fig 1 and 2?

Response: We focused on predictive factors, including the CRP/ALB ratio, for EAD occurrence. We also compared postoperative outcomes, such as overall patient survival, between patients with and without EAD. Because of the retrospective study design, we were not able to analyze the causes of patient mortality in detail, so we focused on the prevalence of all-cause patient mortality. On re-analysis, because re-liver transplantation was sometimes performed due to graft failure, we investigated the incidence of re-transplantation as a graft failure marker (Table 5). We have deleted the redundant Figures 1 and 2 and added the incidence of re-transplantation due to graft failure and overall patient mortality using the χ2 test or Fisher’s exact test, as appropriate, in Table 5 (pages 25 - 26).

4. Why could the authors find donor factors being the risk for EAD? We could not simply accept only one recipient preoperative parameter can be indicator for EAD and graft loss (including long-term patient survival). They should refer this point at discussion, not only one sentence as a limitation since many previous articles focused on donor factors in LDLT.

Response: Donor-graft factors play a critical role in EAD occurrence after liver transplantation. We analyzed the relationship of donor-graft characteristics, such as age, sex, BMI, graft-recipient weight ratio, graft ischemic time, and fatty change, with EAD occurrence (Tables 2 and 4; pages 17, 18 and 24). A multivariate analysis showed that increased donor age had a negative impact on early postoperative graft recovery. Previous LDLT studies suggested that grafts from younger donors had better vascular resistance, higher compliance of the hepatic parenchyma, and more favorable regeneration than those from older donors. Consequently, donor age was a critical and independent factor in recipient survival (Discussion; page 30). 

 Our study population consisted of patients who underwent LDLT. Because there are characteristic differences between LT from living (i.e., smaller but healthy) and deceased (i.e., larger but with some injury) donors, the impact of donor-graft factors on EAD occurrence may be not comparable between both types of LT. Therefore, additional studies are required to validate the predictive role of the CRP/ALB ratio for EAD in LT not only from living donors, but also from deceased donors (Limitations; page 31). 

5. As an objective parameter, “intraoperative portal pressure (or portal flow) after reperfusion “ is very important since it is well-known factor for developing “small for size syndrome”.

Response: We investigated intraoperative data on the hepatic arterial resistive index and portal flow. The Doppler ultrasonographic findings (i.e., hepatic artery resistive index and portal venous flow) were comparable between the non-EAD and EAD groups (Table 2; page 17), and there was no statistical association of these Doppler findings with EAD occurrence (Table 4; page 24). Our results disagree with those of previous studies that identified an association between hepatic flow and EAD occurrence. This may be because of the regular assessment of hepatic vascular flow and appropriate implementation of surgical modulation during surgery. During LDLT, after completion of the hepatic vascular and ductal anastomoses (hepatic vein, portal vein, hepatic artery, and bile duct in a serial manner), the patency of the hepatic vascular flow (portal venous flow and hepatic arterial resistive index) was assessed using Doppler ultrasonography (Prosound SSD-5000; Hitachi Aloka Medical, Tokyo, Japan). Surgical inflow modification using splenectomy, splenic artery ligation, or a portocaval shunt was performed based on the discretion of the transplant surgeons (Patients and Methods; page 8). 

6. In addition to EAD, how was the postoperative complications, such as ascites, infection, etc (makers for small for size syndrome) in high CRP/alb ?

Response: We compared the postoperative complications, including ascites and infection, between patients with a CRP/ALB ratio ≤ and > 20%. Patients with a CRP/ALB ratio > 20% had a longer duration of hospitalization and higher incidences of poor kidney function, infection, and re-transplantation due to graft failure, as well as higher overall mortality, than those with a CRP/ALB ratio ≤ 20% (S1 Table; page 25).

S1 Table. Postoperative outcomes of patients with CRP/ALB ratios ≤ and > 20% 

Group CRP/ALB ratio ≤ 20% CRP/ALB ratio > 20% p

n 340 256 

Hospital stay (days) 25 (21–34) 28 (21–43) 0.006

During follow-up period 

 eGFR-based kidney function grade < 0.001

 Stage 1 263 (77.4%) 144 (56.3%) 

 Stage 2 50 (14.7%) 44 (17.2%) 

 Stage 3a 11 (3.2%) 23 (9.0%) 

 Stage 3b 11 (3.2%) 31 (12.1%) 

 Stage 4 3 (0.9%) 10 (3.9%) 

 Stage 5 2 (0.6%) 4 (1.6%) 

 Ascites 107 (31.5%) 88 (34.4%) 0.454

 Infection 25 (7.4%) 32 (12.5%) 0.034

 Graft rejection 72 (21.2%) 44 (17.2%) 0.223

 De novo cancer occurrence 26 (7.6%) 18 (7.0%) 0.776

 Re-transplantation 5 (1.5%) 19 (7.4%) < 0.001

 Overall patient mortality 35 (10.3%) 55 (21.5%) < 0.001

Abbreviations: EAD, early allograft dysfunction; eGFR, estimated glomerular filtration rate.

NOTE: Values are expressed as the median (interquartile) and number (proportion). 

7. Since CRP/alb is preoperative parameter, if they find CRP/alb>20, what do they do? Do they treat them with antibiotics? Have the patinet better nutritional condition before preceding to transplant? They should discuss this point.

Response: Because chronic inflammation was potentially associated with cirrhosis and damage-associated molecular patterns may originate from hepatocytic injury, including sterile inflammation, our study excluded patients with preoperative signs of infection. Previous studies have shown that in patients with hepatitis B virus (HBV)-related decompensated cirrhosis, antiviral treatment leads to significant downregulation of the MELD score; it is now generally recognized that antiviral therapy can improve the prognosis of HBV-related cirrhosis. An HBV study by Huang et al. (reference No. 53; page 41 - 42) suggested that the CRP/ALB ratio was also significantly downregulated in response to antiviral therapy. This suggests that antiviral therapy influences the predictive capacity of the CRP/ALB ratio. Further research is required to investigate the dose and timing of antiviral therapy in other types of hepatitis (Discussion; pages 29 - 30). 

Minor

1. Table 1, they should stratify BMI because we don’t believe BMI will show liner relationship with EAD.

Response: As per the reviewer’s comments, we compared the stratified BMI values between patients with and without EAD (Table 1; page 14).

Reviewer #2: This study was performed to determine the association between the ratio of C-reactive protein to albumin (CRP/ALB) and the risk of early allograft dysfunction(EAD) in patients undergoing living donor liver transplantation (LDLT).The easily calculated CRP/ALB ratio may allow estimation of the risk of EAD after LDLT and can provide additional information that may facilitate overall estimation of the patient’s condition.

The manuscript is written well but there are several concerns.

comments.

1. In table 3, the predictive accuracy of CRP/ALB was found to be higher than that of other inflammatory markers, such as CRP, ALB, WBC count, and NLR. But in table 4, among inflammatory markers, CRP only nor ALB are not listed. The authors should list up CRP and ALB independently.

Response: As per the reviewer’s comments, we added CRP and albumin to Table 4 (page 23). We had analyzed the association of clinical factors, including C-reactive protein and albumin, with EAD occurrence, and found that the CRP/ALB ratio was an independent predictor. Because of the potential for inter-correlations among CRP, ALB, and CRP/ALB ratio, we included only CRP/ALB ratio in the previous version of Table 4.

2. In table 5, Patients with EAD had a longer duration of hospitalization and higher incidence rates of poor kidney function and infection than those without EAD. These postoperative conditions such as poor kidney function and infection cause EAD? If these postoperative parameters included in multivariable logistic regression, how about significancy?

Response: As per the reviewer’s comments, we analyzed the correlation between postoperative kidney function, infection, and EAD occurrence (Table A), as well as the association between clinical factors, including postoperative kidney function and infection, and EAD occurrence (Tables B and C).

Table A. Correlation of postoperative kidney function, infection, and early allograft dysfunction

 eGFR-based kidney function grade Infection

Early allograft dysfunction 0.264** 0.201**

**p=0.01 using Spearman’s method

Table B. Association of postoperative kidney function and infection with EAD occurrence in a univariate logistic regression analysis

 β Odds ratio 95% CI p

eGFR-based kidney function grade 0.560 1.751 1.468–2.089 < 0.001

Infection 1.420 4.136 2.255–7.586 < 0.001

Abbreviation: eGFR, estimated glomerular filtration rate.

Table C. Association of clinical factors, including postoperative kidney function and infection, with EAD occurrence based on a multivariate logistic regression analysis. Potentially significant factors (p < 0.1) identified in the univariate analysis were entered into multivariate forward and backward logistic analyses.

 Multivariate logistic regression analysis

 β Odds ratio 95% CI p

Preoperative recipient finding 

MELD score (point) 0.048 1.049 1.022–1.078 < 0.001

CRP/ALB (%) 0.004 1.004 1.001–1.007 0.012

Intraoperative recipient finding 

Fresh frozen plasma (units) 0.038 1.038 1.001–1.077 0.041

Donor-graft finding 

Age (years) 0.025 1.025 1.001–1.049 0.038

Postoperative recipient finding 

eGFR-based kidney function grade 0.269 1.308 1.054–1.624 0.015

Infection 0.832 2.299 1.055–5.010 0.036

AUC=0.799; 95% CI=0.742–0.856; p < 0.001 in the predictive model.

 Postoperative kidney function and infection were closely related to EAD occurrence. However, we were not able to determine whether postoperative kidney function and infection were independent predictors of EAD occurrence since they may share preoperative and intraoperative risk factors, such as blood product transfusion. A previous study by Wadei et al. (reference No. 47; pages 40 - 41) suggested that EAD after LT was associated with short- and long-term kidney function impairment. They found that EAD development causes oxidative stress due to the generation of reactive oxygen species and initiates a systemic inflammatory response that extends beyond the hepatic allograft. Thus, renal injury was a direct effect of the circulating cytokines or secondary to the systemic inflammatory response created by the EAD milieu, with the subsequent development of AKI leading eventually to end-stage renal disease. Therefore, further research is required to elucidate the effect of the postoperative development of AKI and infection on early graft function recovery in the liver transplantation setting.

We thank you and the reviewers again for your extremely valuable and insightful comments. We believe that our manuscript has been improved as a direct result of the review process. We hope that our revised manuscript is now deemed suitable for publication in PLoS ONE.

Sincerely,

Min Suk Chae, MD, PhD

---

## [Decision Letter · Decision Letter 1]

26 Nov 2019

Predictive utility of the C-reactive protein to albumin ratio in early allograft dysfunction in living donor liver transplantation: A retrospective observational cohort study

PONE-D-19-22048R1

Dear Dr. Chae,

We are pleased to inform you that your manuscript has been judged scientifically suitable for publication and will be formally accepted for publication once it complies with all outstanding technical requirements.

With kind regards,

Yun-Wen Zheng

Academic Editor

PLOS ONE

Additional Editor Comments (optional):

Reviewers' comments:

Reviewer's Responses to Questions

**Comments to the Author**

1. If the authors have adequately addressed your comments raised in a previous round of review and you feel that this manuscript is now acceptable for publication, you may indicate that here to bypass the “Comments to the Author” section, enter your conflict of interest statement in the “Confidential to Editor” section, and submit your "Accept" recommendation.

Reviewer #1: All comments have been addressed

Reviewer #2: All comments have been addressed

2. Is the manuscript technically sound, and do the data support the conclusions?

Reviewer #1: Yes

Reviewer #2: Yes

3. Has the statistical analysis been performed appropriately and rigorously? 

Reviewer #1: Yes

Reviewer #2: Yes

4. Have the authors made all data underlying the findings in their manuscript fully available?

Reviewer #1: Yes

Reviewer #2: Yes

5. Is the manuscript presented in an intelligible fashion and written in standard English?

Reviewer #1: Yes

Reviewer #2: Yes

6. Review Comments to the Author

Reviewer #1: The paper “Predictive utility of the C-reactive protein to albumin ratio in early allograft dysfunction in living donor liver transplantation: A retrospective observational cohort study”. is now properly revised. The authors have answered all our comments wisely. We could not find any issues now.

Reviewer #2: The revision is well written and all of the comments were replied well. The revised version is acceptable.

7. PLOS authors have the option to publish the peer review history of their article (what does this mean?). If published, this will include your full peer review and any attached files.

Reviewer #1: No

Reviewer #2: Yes: Soichiro Murata

---

## [Editor Report · Acceptance letter]

3 Dec 2019

PONE-D-19-22048R1 

Predictive utility of the C-reactive protein to albumin ratio in early allograft dysfunction in living donor liver transplantation: A retrospective observational cohort study 

Dear Dr. Chae:

I am pleased to inform you that your manuscript has been deemed suitable for publication in PLOS ONE. Congratulations! Your manuscript is now with our production department. 

With kind regards,

on behalf of

Dr. Yun-Wen Zheng 

Academic Editor

PLOS ONE